# Link between Energy Investment in Biosynthesis and Proteostasis: Testing the Cost–Quality Hypothesis in Insects

**DOI:** 10.3390/insects14030241

**Published:** 2023-02-28

**Authors:** Taiwo Iromini, Xiaolong Tang, Kyara N. Holloway, Chen Hou

**Affiliations:** 1Department of Biological Sciences, Missouri University of Science and Technology, Rolla, MO 65409, USA; 2Department of Animal and Biomedical Sciences, School of Life Sciences, Lanzhou University, Lanzhou 730020, China

**Keywords:** life history, energy, growth, somatic maintenance, proteasome

## Abstract

**Simple Summary:**

The energy cost for synthesizing bio-tissue varies greatly among different species. We hypothesize that high energy cost stems from maintaining the protein homeostasis (proteostasis), including the cost of unfolding, degrading, and resynthesizing the proteins that have errors; as such, a large amount of synthesized protein does not contribute to growth. Our hypothesis predicts that the species with high energy cost have better cellular viability under stress due to better proteostasis, higher proteasome activities that break down proteins, and slower growth for a given amount of RNA that synthesizes proteins. We tested the hypothesis on two insect species, painted lady caterpillar and Turkestan cockroach nymph, the former’s biosynthetic energy cost being 20 times lower than the latter. The results support our hypothesis.

**Abstract:**

The energy requirement for biosynthesis plays an important role in an organism’s life history, as it determines growth rate, and tradeoffs with the investment in somatic maintenance. This energetic trait is different between painted lady (*Vanessa cardui*) and Turkestan cockroach (*Blatta lateralis*) due to the different life histories. Butterfly caterpillars (holometabolous) grow 30-fold faster, and the energy cost of biosynthesis is 20 times cheaper, compared to cockroach nymphs (hemimetabolous). We hypothesize that physiologically the difference in the energy cost is partially attributed to the differences in protein retention and turnover rate: Species with higher energy cost may have a lower tolerance to errors in newly synthesized protein. Newly synthesized proteins with errors are quickly unfolded and refolded, and/or degraded and resynthesized via the proteasomal system. Thus, much protein output may be given over to replacement of the degraded new proteins, so the overall energy cost on biosynthesis is high. Consequently, the species with a higher energy cost for biosyntheses has better proteostasis and cellular resistance to stress. Our study found that, compared to painted lady caterpillars, the midgut tissue of cockroach nymphs has better cellular viability under oxidative stresses, higher activities of proteasome 20S, and a higher RNA/growth ratio, supporting our hypothesis. This comparative study offers a departure point for better understanding life history tradeoffs between somatic maintenance and biosynthesis.

## 1. Introduction

Life history theory predicts tradeoffs between production of new tissues and somatic maintenance when available resources and energy are limited [1]. Previous empirical and theoretical studies have achieved a considerable amount of knowledge of how the energy allocation strategies of species with different life histories constrain their development, reproduction, and senescence process [2,3,4,5,6]. However, most of the life history studies, theoretical or empirical, have failed to pay enough attention to an important energetic trait—the energy required to synthesize one unit of biomass, denoted as *E*_m_ hereafter. *E*_m_ is the indirect cost of growth, not deposited in the biomass. Instead, it is a part of respiration, and dissipated as heat, in contrast to the combustion energy content of tissues. It mainly includes the metabolic work required to assemble the monomers to polymers, fold them, and transport them to the required location [7,8]. *E*_m_ plays an important role in life history, as it is directly related to the ontogenetic growth rate [9,10,11,12,13,14]. If biosynthesis is cheap, i.e., low *E*_m_, growth will be faster. Moreover, our recent study [11] suggested that *E*_m_ may also partly contribute to the cellular resistance to oxidative stress.

Since *E*_m_ has been considered a fundamental biochemical property of cells [7,15], all the theoretical studies, to the best of our knowledge, such as the metabolic theory of ecology [9,16] and the dynamic energy budget theory [17,18], treated it as a constant across species. Wieser [14] and Ricklefs [12] concluded that the “consensus value” of *E*_m_ is about 7.2 KJ/g of dry body mass (KJ/gdbm) for a wide range of organisms. Previous comparative studies have found *E*_m_ varying within a four-fold range (e.g., see [14,19,20] and Figure 1) across ectothermic and endothermic species with very different body sizes and biomass components, including fish, amphibians, reptiles, Bivalvia, birds, and mammals. The average of the above species is 9.16± 4.30 KJ/gdbm, close to the “consensus value” of 7.2 KJ/gdbm. We have found [11,21], however, a great variation in *E*_m_ in insect species. Painted lady caterpillars (*Vanessa cardui*) and Turkestan cockroach nymphs (*Blatta lateralis*), which have a similar dry body mass range, have sharply different values of *E*_m_, i.e., 0.336 and 6.91 KJ/g of dry mass (gdm), respectively. The variation is not caused by diet and activity levels of the animals, and cannot be explained by the difference in their body components [11].

The *E*_m_ of cockroach nymphs is similar to that of the other species listed in Figure 1, including fish, birds, and some mammals. However, the value of the painted lady caterpillar is 27 times lower than the average, and 20 times smaller than that of cockroaches. We suggest that the difference in *E*_m_ may be understood from the viewpoint of the life histories of holometabolous and hemimetabolous insects. The painted lady caterpillar grows almost 30-fold faster than the cockroach nymph with the same dry body mass range [11]. In the former, a large portion of biomass synthesized during the larval stage serves as energy storage for reproduction, instead of functional structures, during the adult stage. Such tissues are disintegrated and remodeled during the pupal stage [28,29,30,31,32]. Moreover, the larvae of many holometabolous insects need to reach a critical weight for successful pupation [33,34], and insects’ fecundity is positively correlated to body size [35], both of which require fast growth. Considering the tissue disintegration during metamorphosis, the pressure for fast larval growth, and the short adult lifespan, synthesizing high quality bio-tissues during the larval stage, which requires a high amount of energy, would be economically wasteful for these species.

Physiologically, we have proposed a “cost-quality hypothesis” to explain the difference in *E*_m_ [11]. The cost of biosynthesis determines the “cellular quality” of bio-tissues, including the number of errors in protein and DNA sequences, resistances to stress, and the rate of senescence. Taking protein homeostasis as an example, the value of *E*_m_ depends on amino acid compositions [36,37,38], which affect the protein stabilities [39,40] and the proofreading efforts [41,42] that are tightly associated with protein fidelity [43]. So, we hypothesize that the value of *E*_m_ is largely determined by the tolerance to mistakes in protein synthesis. Newly synthesized proteins have a high chance of misfolding and aggregation, which will result in protein toxicity [44]. Cells possess a protein quality control mechanism to maintain the integrity of the proteome (proteostasis) [45], refolding, degrading, and sequestering misfolded polypeptides [46,47]. One of the major arms of the quality control system is the ubiquitin-proteasomal system (UPS) [44,48,49,50,51], which efficiently degrades protein waste, and one of the major components of the system is 20S proteasome [52,53]. A species with low error tolerance would spend more energy (high *E*_m_) on making one unit of protein, if newly synthesized proteins that contain intolerable errors are degraded and resynthesized via the proteasomal activities. These activities, on one hand, slow down the net gain of biomass, and therefore increase the value of *E*_m_; on the other hand, they also slow down protein aggregation, and improve protein homeostasis [54,55,56,57]. Our goal of this study was to test this hypothesis, and investigate the relationship between E_m_ and cellular resistance, using two insect species, the caterpillar of the painted lady (*Vanessa cardui*) and the nymph of the Turkestan cockroach (*Blatta lateralis*).

Our hypothesis makes three testable predictions. First, species with higher *E*_m_ have better tissue quality, and higher cellular resistance to oxidative stress. Under stress, cells will first initiate a series of defense and repairing mechanisms. If the stressful stimuli continue, cells will activate death signaling pathways [58]. To test this prediction, we induced oxidative stress by tert-butyl hydroperoxide (t-BHP) at several concentrations, and assayed the cell viability and percentage of apoptotic cells from the midguts of painted lady caterpillars and cockroach nymphs. Second, species with higher *E*_m_ have higher proteasomal activities, which cost a considerable amount of ATP and directly determine protein homeostasis [54,55,56,57]. We measured proteasomal activities in the midgut tissues of painted lady caterpillars and cockroach nymphs to test this prediction. Finally, species with higher *E*_m_ values spend a large percentage of protein on replacing the newly synthesized proteins that have errors. Thus, for a given amount of RNA, the high-*E*_m_ species deposit a smaller fraction of newly synthesized protein as new bio-tissue than the low-*E*_m_ species, leading to a higher RNA-to-growth ratio (RNA:growth) [59,60]. We measured the RNA content and biomass growth rate of the midgut tissue of these species to test this prediction.

## 2. Materials and Methods

### 2.1. Animal Rearing and Sampling

Cockroach nymphs and painted lady caterpillars were reared at 25 ± 1 °C on a long day cycle (17 h light: 7 h dark), except the samples for the proteasomal activities assays were reared at 22 and 30 °C. Butterfly caterpillars were fed ad libitum with sucrose and a protein-based diet (Carolina Biological Supply, NC. 80% moisture; per unit of dry food has 13–15% of protein content and a negligible amount of lipid content). Cockroaches were supplied with Wardley Pond Pellets (Hartz Mountain Corp., Secaucus, NJ, USA; the protein and lipid contents of the dry mass are 33% and 5.5%, respectively). Water supply was unlimited. The caterpillar samples were chosen in instars 4 and 5, the last two instars of their development, as they were still growing but large enough to easily dissect. Samples with any sign of pupation were discarded. The cockroach samples were chosen so that the body mass range was similar in both species. The reason we chose similar body size range is that the qualities of interest that we tried to compare in this study, such as proteasomal activity and RNA content, are related to growth rate, and growth rate is linearly correlated with body mass in these two species (see below).

The insects were cleaned via submersion in a set of three surface sterilizing solutions for 2 min each. The cleaning solutions included a methyl-4 detergent solution, a 20% septisol solution, and a 1% bleach solution, to remove any contaminants clinging to the insects themselves. The insects were then taken to a sterile hood, where their midguts were removed, rinsed, and placed into pre-weighed centrifuge tubes. Upon successful acquisition of the organ, it was immediately placed on ice. It was then weighed and partitioned into portions for extractions. Any spare tissue was frozen and stored at −80 °C. Once the organ was removed, it was split down the middle and cut into several sections. These pieces were then vortexed in phosphate buffered saline to remove food contaminants.

### 2.2. Cell Sampling and Culture

Midgut cells were harvested using a modified protocol [61] from animals that were in the last two instars of their developmental period. We followed the published protocol for insect cells [62] to maintain the primary cell culture. Briefly, midguts were placed in insect physiological solution (NaCl 178 mM, KCl 4.3 mM, CaCl_2_ 4.3 mM, NaHCO_3_ 3.8 mM, 0.5% gentamicin, 0.01% antibiotic antimycotic PH 6.5) and washed twice before transferring to a well on a 6-well plate. Cells were maintained in Grace Insect Medium (Thermo Fisher Scientific, Rockford, IL, USA) supplemented with 10% heat-inactivated fetal bovine, 0.1% gentamicin, vitamin mixture, and 0.1% antibiotic antimycotic at 25 °C. After 24 h, primary cell culture was filtered using 70 µm cell strainers (CLS431751, Thermo Fisher Scientific, Rockford, IL, USA) to remove gut explants. Cells were collected after gentle pipetting and washed twice in 0.1 M cold PBS buffer (8.00 g NaCl, 0.20 g KCl, 1.29 g Na_2_HPO_4_·3H_2_O, 0.20 g KH_2_PO_4_, 1000 mL ddH_2_O, pH 7.4) to be used in further analysis.

### 2.3. Cell Viability and Apoptosis

tert-butyl hydroperoxide (t-BHP), which is more thermodynamically stable than H_2_O_2_, was used to induce oxidative stress. Cells were collected after gentle pipetting and an initial cell density of 1 × 10^6^ cells mL^−1^ was seeded on a six-well plate. After 30 min, fresh media containing t-BHP was added to final concentrations of 3, 6, 9, 12, 15, 50, 100, and 200 mM. After a 6 h incubation, treated cells were collected directly in the 15 mL centrifuge tubes and washed twice at room temperature in 0.1 M cold PBS buffer. Cell pellets were resuspended in 5 µL 7AAD staining solution, incubated in the dark at room temperature for 15 min, and cells were analyzed within an hour. Using the flow cytometer (Beckman Coulter Cytoflex, Indianapolis, IA, USA), Forward scatter (FSC) vs. Side scatter (SSC) gates were set appropriately to exclude debris and cell aggregates, untreated cells (negative control) stained with 7-AAD were used to define the basal level of dead cells and set up the necessary laser compensation, and the 7-AAD fluorescence was collected at the FL3 channel.

Simultaneous staining of cells with Annexin V–FITC (green fluorescence) and 7-aminoactinomycin (7AAD) (red fluorescence) allows for the discrimination of intact cells (Annexin V–FITC negative, 7AAD negative), early apoptotic cells (Annexin V–FITC positive, 7AAD negative), late apoptotic cells (Annexin V–FITC positive, 7AAD positive), and dead cells (Annexin V–FITC negative, 7AAD positive). Negative controls with untreated cells were used to define the basal level of apoptotic and necrotic cells. Flow cytometer (Beckman Coulter Cytoflex) FSC vs. SSC gates was set to exclude debris and cell aggregates. Single-color controls were used to set up the necessary laser compensation. The measurements at 0, 12, and 15 mM t-BHP were repeated three times, and the other treatments were performed once.

### 2.4. Proteasome Activity

The insects were anesthetized with carbon dioxide, and the midguts were taken and rinsed with distilled water. The midguts were weighed and homogenized by adding phosphate buffer (137 mM NaCl, 2.7 mM KCl, 8.1 mM Na_2_HPO_4_, 1.5 mM KH_2_PO_4_, pH 7.2–7.4) at a ratio of 1:9. The homogenate was then centrifuged at 4 °C, 3000 rpm for 10 min and the supernatant for the determination of proteasome activities. Protein concentration was determined by Coomassie Brilliant Blue G-250 (CBB, Bio-Rad, Hercules, CA, USA).

Proteasome measurement was conducted according to the published protocols of Breusing, Nicolle et al. [63] and Zeng et al. [64], with some modifications. The activity of proteasome in midguts was assessed using the fluoropeptide Suc-Leu-Leu-Val-Tyr-AMC (chymotrypsin-like), and MG132 was used as a proteasome inhibitor [65]. For each sample, the assay was performed in AB buffer (20 μL of DMSO to each 1 mL of PBS) or IB buffer (20 μL of stock solution of MG132 (20 mM in DMSO) to each 1 mL of PBS, with final concentration of the inhibitor at ~35 μM), respectively. The assay contained 20 μL tissue homogenate,170 μL AB/IB, and 10 μL fluorescent protein AMC, and the mixture was incubated for 30 min at 37 °C and the fluorescence determination was continuously performed at 360 nm excitation and 485 nm emission in a Microplate Spectro fluorometer (BMG Labtech Fluostar Omega, Offenburg, Germany). The slope obtained in the presence of MG132 was subtracted from the non-presence of MG132 to eliminate the proteasome non-specific activity. Proteasomal activity was determined as the increase in fluorescence of the reaction products and expressed as fluorescence units (FU)/min/mg protein. All the assays were conducted at 25 °C, although two groups of each species were reared at 22 and 30 °C separately.

### 2.5. Protein Extraction and Quantity

I-PER (Thermo Fisher Scientific, St. Louis, MO, USA) was used to extract protein from 1.0–2.5 mg of midgut cells of both species. A protease inhibitor and EDTA were included to stabilize the solution and prevent protein degradation or precipitation. This solution was then incubated on ice for 10 min, and then another 15 min on the centrifuge at 15,000 RCF. The supernatant was pipetted into a separate container. This entire procedure was performed on ice to preserve the protein and further prevent degradation. The Bradford Protein Assay Quant-Ti kit (Invitrogen, Waltham, MA, USA) uses a colorimetric dye that binds to protein. A 96-well plate was used, where each standard was plated 4 times, each sample plated 3 times, and each plate read twice. All of the samples and standards were placed into the well first. Dye was then added. A FLUOStar Omega plate reader (BMG Labtech Fluostar Omega, Offenburg, Germany) was set to a 595 nm wavelength.

### 2.6. RNA Extraction and Quantity

Midgut tissues of both species, around 10 mg, were used for this procedure. Throughout this procedure all surfaces and tools were cleaned with Rnase-zap in order to prevent RNA degradation. The PureLink RNA Extraction kit (Thermo Fisher Scientific, St. Louis, MO, USA) was used for the extraction of the RNA via spin column filtration and washes, resulting in total RNA from the sample suspended in 30 uL of RNase free water. A Quant-iT™ RNA Assay Kit (Invitrogen, Waltham, MA, USA) was used for this procedure. Samples were plated and read in a FLUOstar Omega plate reader (BMG Labtech Fluostar Omega, Offenburg, Germany). Each standard was plated 4 times, each sample was plated 3 times, and each plate was read twice. Data acquired were averaged for each specific sample. The plate was read at 660 nm. Considerations for interactions between components of the assay were not required, as the Broad Range Assay kit and the PureLink Mini RNA Extraction kit were made to operate together. The only item not belonging to a kit in use was the Rnase inhibitor used on surfaces and tools before use, and this item does not interfere.

### 2.7. Data Analysis

Both RNA and protein contents are expressed in the unit of µg/mg of dry midgut mass. Midguts were removed from the body, rinsed with distilled water, and oven-dried at 65 °C for 72 h to determine the dry mass. The rest of the body was dried the same way. The dry mass of the bodies and midguts of all the samples was measured to the nearest 0.1 mg. The whole body growth rates of both species were estimated by the previously obtained relationships between growth rate and body dry mass [11], *G*_body_ = 0.0130 *M*_body_ for cockroaches, and *G*_body_ = 0.354 *M*_body_ for caterpillars, where *G*_body_ is whole body growth rate in the unit of grams of dry mass/day, and *M*_body_ is dry body mass in the unit of grams. We linearly regressed the midgut dry mass to the whole body dry mass, as *M_midgut_* = *a* × *M*_body_, where *a* is the proportional coefficient. This relationship enables us to estimate the growth rate of the midgut as *G_midgut_* = *a* × *G*_body_. We measured RNA and protein concentrations in the midgut tissues of 30 animals in each species. We then calculated the growth-specific RNA content by dividing it by the midgut growth rate of each animal, as RNA/*G*_midgut_. For both proteasome activities and RNA/growth rate, the two groups of samples are independent, so the two-sample *t*-test was used to compare the two species using OriginLab (OriginPro 2023, OriginLab Corporation, Northampton, MA, USA). Before running the *t*-test, we compared the variances between the groups using OriginLab. Since they have different variances (*p* < 0.0001), we chose Welch’s *t*-test.

## 3. Results

### 3.1. Dry Midgut Mass

We found the dry midgut mass to be proportional to the dry body mass of both species, as *y* = 0.055*x* and *y* = 0.041*x* for caterpillars and cockroach nymphs, respectively (Figure 2).

### 3.2. Cell Viability and Apoptosis

We assayed the cell viability and percentage of apoptotic cells from painted lady caterpillar and cockroach midguts under oxidative stress induced by tert-butyl hydroperoxide (t-BHP). Figure 3 shows an example of the flow cytometer reading of cell death under two concentrations of t-BHP. Figure 4 summarizes the percentage of the dead cells under eight concentrations of t-BHP. Caterpillars have higher percentages of apoptotic and dead cells than cockroaches at most concentrations of t-BHP, except the two highest concentrations (100 and 200 mM), which caused extremely low cell viability in both species.

Using Annexin V and 7AAD double staining, the percentages of apoptotic cells at three concentrations of t-BHP were obtained (Figure 5 and Figure 6). At concentrations of 12 and 15 mM, t-BHP induced apoptosis in painted lady caterpillar cells (10.9 ± 0.3% at 12 mM and 12.3 ± 1% at 15 mM t-BHP), but almost no apoptosis in Turkestan cockroach cells (0.25 ± 0.08% at both concentrations). At 0 mM t-BHP, the apoptosis was below 0.23% in both species.

### 3.3. Proteasome Activity

The mean values of proteasome activities in cockroach nymph midgut tissues were 93.1 ± 47.4 (N = 26, median = 79.2, 95% CI: 74.0, 112.3) and 103.8 ± 60.0 (N = 29, median = 103.6, 95% CI: 81.0, 126.6) FU/min/mg protein at 22 and 30 °C, respectively; those of caterpillar midgut tissues were 14.8 ± 11.5 (N = 23, median = 14.6, 95% CI: 9.90, 19.8) and 28.5 ± 18.6 (N = 25, median = 25.2, 95% CI: 20.8, 36.2) FU/min/mg protein at 22 and 30 °C, respectively. Figure 7 shows that the proteasome activities in cockroaches were significantly higher than those in caterpillars (Welch *t*-test, t = 8.16, DF = 28.3, *p* < 6.47 × 10^−9^ at 22 °C, and t = 6.41, DF = 34.1, *p* < 2.48 × 10^−7^ at 30 °C).

### 3.4. RNA/Growth Ratio

The RNA content values in midgut tissues of cockroaches and caterpillars were 6.91 ± 9.73 (N = 30; median = 2.97; 95% CI: 3.28, 10.55) and 5.85 ± 3.40 (N = 30; median = 5.06; 95% CI: 4.58, 7.12) µg/mg, respectively. The protein contents were 608.7 ± 245.9 (N = 30) and 564.4 ± 403.2 (N = 30) µg/mg of dry midgut mass for cockroaches and caterpillars, respectively. The RNA/protein ratio in cockroach midguts (0.0137 ± 0.0216; N = 30; median = 0.0049; 95% CI: 0.00559, 0.0217) was lower, but not significantly (Welch’s *t*-test, *t* = −1.41, DF = 32.7, *p* = 0.169), than that of caterpillars (0.0364 ± 0.0856; N = 30; median = 0.01166; 95% CI: 0.00438, 0.0683).

The RNA/growth ratio of cockroaches, 276.4 ± 318.6/(mg/day) (N = 30; median = 140.4; 95% CI: 157.5, 395.4), was significantly higher (Welch’s *t*-test, t = −4.70, DF = 29, *p* < 0.000058) than that of caterpillars, 3.25 ± 1.63/(mg/day), (N = 30; median = 2.83; 95% CI: 2.64, 3.86) (Figure 8). OriginLab identified three outliers in the cockroach dataset (circled in Figure 8). Removing the outliers gave the mean value of the cockroach RNA/growth ratio 187.2 ± 166.0/(mg/day) (N = 27; median: 119.3; 95% CI: 121.5, 252.9), which was still significantly higher than that of caterpillars (Welch’s *t*-test, *t* = −5.76, DF = 26.0, *p* < 0.0000046).

## 4. Discussion

The variations in the RNA content data are large, which in turn cause large variations in the RNA/protein data. As explained in the Introduction, RNA content varies with tissue growth rate, which in turn varies with body mass. As shown in Figure 2, the body masses of the two species used in this study have a 5~6-fold variation. This partially explains the variation in RNA content. However, the RNA/growth ratio, which is a normalized quantity (normalized with respect to growth), should vary in a narrow range. The OriginLab software identified three outliers out of 30 points in the cockroach dataset (circled in Figure 8). Removing the outliers makes the mean value of RNA/growth ratio of cockroaches smaller than the mean with the outliers, but still significantly higher than that of caterpillars.

The induced oxidative stress assays show that at six levels of t-BPH concentrations (3, 6, 9, 12, 15, 50 mM) the cockroach midgut cells have higher cellular viability than caterpillars (Figure 3 and Figure 4). Two high levels of t-BPH (100 and 200 mM) caused extreme cell death in both cockroaches and caterpillars; thus, the difference between the two species was not obvious. The apoptosis assays also show (Figure 5 and Figure 6) that caterpillar midgut cells are more vulnerable to oxidative stress than cockroach cells.

Cellular viability directly relies on proteostasis, the loss of which is a hallmark of cellular aging [66,67,68,69]. Cells have a series of proteostasis machinery. Together, with the autophagy-driven vacuolar proteolysis and endoplasmic-reticulum-associated degradation [70], two major arms of the quality control system are the molecular chaperon and the ubiquitin-proteasomal system (UPS) [44,48,49,50,51]. The former identifies the misfolded proteins, promotes refolding, and assists degradation via the UPS [71], while the latter degrades protein waste. As a preliminary test of our hypothesis, we measured the 20S proteasome activities, a major component of UPS, in the midgut cells of two species. The results (Figure 7) show that the midgut cells from cockroaches have significantly higher proteasome activities than those of caterpillars. The high proteasome activities not only partially enable the higher cellular resistance to oxidative stress in the cockroach, which contributes to the results in Figure 3, Figure 4, Figure 5 and Figure 6; equally importantly, they may also partially explain the high biosynthetic energy cost (*E*_m_) in cockroaches, because maintaining proteasome activity is an energetically costly process [48,53,65,66,72]. It is impossible to estimate quantitatively how much of the difference in the values of *E*_m_ between the cockroach and caterpillar can be attributed to the difference in the proteasome activities, because the quantitative measurements of the energetic cost in the activities are not available. Nonetheless, it has been found that protein degradation may cost 20% of the mammalian total energy expenditure [73], so it is safe to assume that the proteasome activities constitute a large fraction of *E*_m_. To fully understand the higher cellular resistance and the associated higher *E*_m_ in cockroaches, more comprehensive quantitative assays on other major arms of cellular homeostasis machinery, such as vacuolar proteolysis and chaperon systems, should be performed. However, those assays are beyond the scope of this preliminary study.

It was found that, within one species, the proteolytic activity increases with cellular growth rate [69,74]. However, our study shows that, across species, compared to the cockroach, the butterfly caterpillar grows much faster, but has lower proteasomal activity. Moreover, it was suggested that the low protein synthesis results in low error protein synthesis, which was proposed to explain the lifespan-extending effect of diet protein restriction [75]. The protein content in the midgut tissue is almost the same in cockroaches and caterpillars (608.7 and 564.4 µg/mg, respectively), but our previous study [11] shows that, in the whole body, the overall protein content is four times higher in caterpillars (482.7 vs. 112.7 µg/mg). So, if everything is the same, the high protein content in caterpillars would indicate a high error protein production, which in turn would result in high proteolytic activities. However, our result shows that the proteasomal activity in cockroaches is 4~5 fold of that in caterpillars (Figure 7). The cost–quality hypothesis proposed here offers an explanation from the angle of life history: Due to the different life history explained in Introduction, caterpillars have a set of “cheaper” cellular apparatus for proteostasis than cockroaches.”

High proteasome activities lead to a high protein turnover rate [48,53,76]. This means that a large amount of newly synthesized protein output from ribosomes is degraded and recycled for replacing the existing protein; it is not deposited as new biomass. Thus, for a given amount of RNA and a given amount of protein output, species with higher proteasome activities (such as cockroaches, compared to caterpillars) will have a lower overall growth rate (body mass gain per unit of time). This is what was observed in this study. We found no significant difference in RNA/protein ratio between cockroaches and caterpillars, but the mean value of the RNA/midgut growth ratio of cockroaches was 85 (=276.4/3.25) times higher than that of caterpillars (Figure 8); even when the three outliers in the cockroach dataset were removed, the mean value was still 58 (=187.2/3.25) times higher. A possible explanation is that the protein levels we saw, which were similar in two species, are merely snapshots; in the cockroach, the proteins measured include those that are malfunctional, unfolded, being degraded, and to be recycled, and the protein synthesized into new biomass in the midgut is only a small fraction, so the overall growth rate of the cockroach is smaller than that of the caterpillar.

The results in Figure 8 echo the recent development in the field of biological stoichiometry. It has long been hypothesized (Growth Rate Hypothesis) that fast-growing organisms need relatively more phosphorus-rich RNA to support rapid rates of protein synthesis, and therefore, growth rate is positively associated with RNA content [59,60,77,78]. While the Growth Rate Hypothesis (GRH) has gained support from observations of a wide range of organisms, the positive association between growth rate and RNA content is often decoupled in many organisms [59,60]. Among other reasons, recent studies suggest [59,60] that the decoupling is due to GRH’s assumption of a constant protein retention, i.e., a fixed fraction of synthesized protein accumulates in the cell, contributing to growth, and that “high rates of protein turnover keep proteins from accumulating in cells, thus decoupling growth from RNA.” Our results, in agreement with this explanation, paint a coherent picture: Compared to the caterpillar, the higher proteasome activities (Figure 7) in the cockroach lead to a higher protein turnover rate. This enables proteostasis and cellular resistance to oxidative stress (Figure 3, Figure 4, Figure 5 and Figure 6) and a higher RNA/growth ratio (Figure 8). Altogether, these traits contribute to the higher biosynthetic energy cost, *E*_m_, in the cockroach. We do admit, however, the results in Figure 7 and Figure 8 only offer indirect explanations for the decoupling of RNA and growth rate. We call for future research on protein turnover and fate for more direct evidence.

We postulate that the physiological differences observed in this study come from the differences in life history between holometabolous and hemimetabolous insects. However, we must acknowledge that our comparative analysis study is narrow, and is confounded by factors other than the differences in life history. For example, cockroach nymphs are able to move to new environments in response to the changes in temperature, moisture, or food availability, but butterfly caterpillars’ capability of moving is limited. In this study, both species were reared in an artificial and benign environment, but in the field they may have different external mortality rates, since cockroaches are able to adapt to a larger variety of conditions, compared to hornworms. All of these differences may affect the energy allocation to biosynthesis, and in turn affect the cellular resistance to stress. Thus, fully testing our life history postulation requires more studies on a broader range of holometabolous and hemimetabolous species with different habitats and traits, such as fecundity, body size, and generation time. Nonetheless, there is a hint from hornworms (*Manduca sexta* larva), a holometabolous species, whose *E*_m_ is only 1.3 KJ/g of dry mass (Figure 1), which is similar to that of the painted lady caterpillar but much smaller than the organisms that have similar life histories to the cockroach. It is worth mentioning that the *E*_m_ that we focused on is the energy required to synthesize biomass during development. The energy for structural disintegration and making glycogen and lipids during the pupal stage is not included. However, making and maintaining glycogen and lipid storage do not require as much energy as protein [7]. This implies an even lower *E*_m_ for the caterpillar during the pupal stage, compared to the cockroach, which does not experience this stage. Further research is required to confirm whether this is true. If proven, it will further confirm the idea that the life history of holometabolous insects results in a cheap energy requirement for biosynthesis.

This study offers a departure point for a better understanding of the life history tradeoff between somatic maintenance and biosynthesis. It suggests that materials that are cheap to synthesize deteriorate faster, and the life history of the species determines whether to make cheap or expensive materials. Thus, the capability of maintaining homeostasis not only depends on the amount of energy allocated to somatic maintenance, but also depends on the energy allocated to biosynthesis, i.e., the quality of the tissue. The significance of the energy requirement for biosynthesis (*E*_m_) in the cellular stress response has been largely ignored in the past. Since previous empirical data only showed slight variation [12,14,20], researchers considered *E*_m_ to be roughly constant across species [9,16,17,18,79]. Therefore, it was thought that *E*_m_ does not contribute to the variations in life history traits, nor to the variations in cellular endurance. Our results highlight *E*_m_’s roles in somatic maintenance, and open a door to a future cellular biology research area: What are the mechanisms that determine the value of *E*_m_, and can we manipulate them for better somatic maintenance?

## Figures and Tables

**Figure 1 insects-14-00241-f001:**
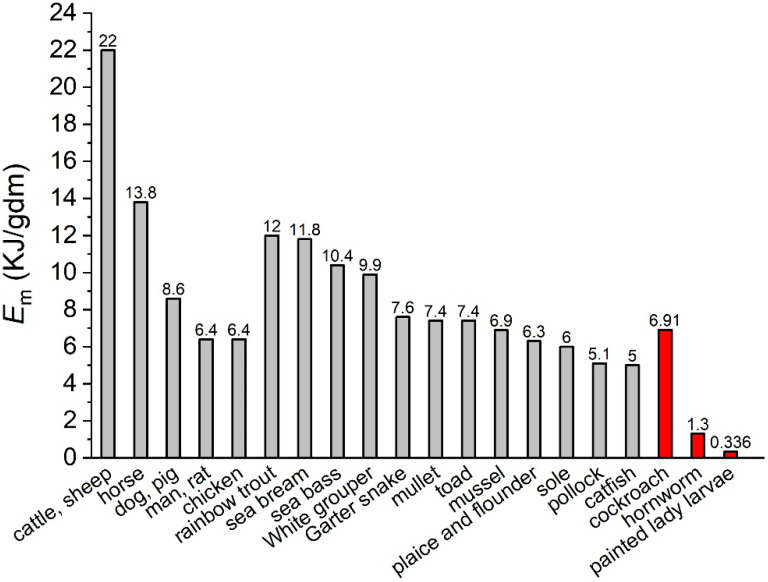
Values of *E*_m_ of a few ectothermic species. (Data collected from [11,13,14,20,21,22,23,24,25,26,27]).

**Figure 2 insects-14-00241-f002:**
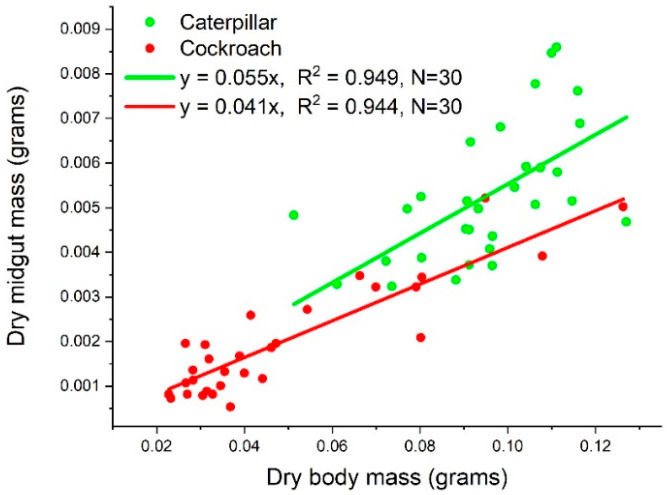
The dry midgut mass as linear functions of body mass of both species.

**Figure 3 insects-14-00241-f003:**
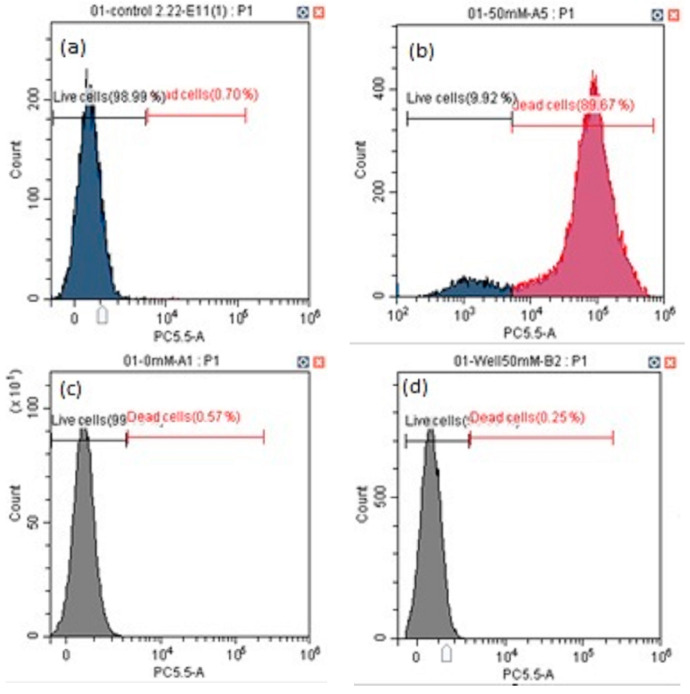
Examples of flow cytometry analysis of cell viability of painted lady caterpillars’ cells (**a** = control, **b** = 50 mM t-BHP) and cockroach nymph cells (**c** = control, **d** = 50 mM t-BHP).

**Figure 4 insects-14-00241-f004:**
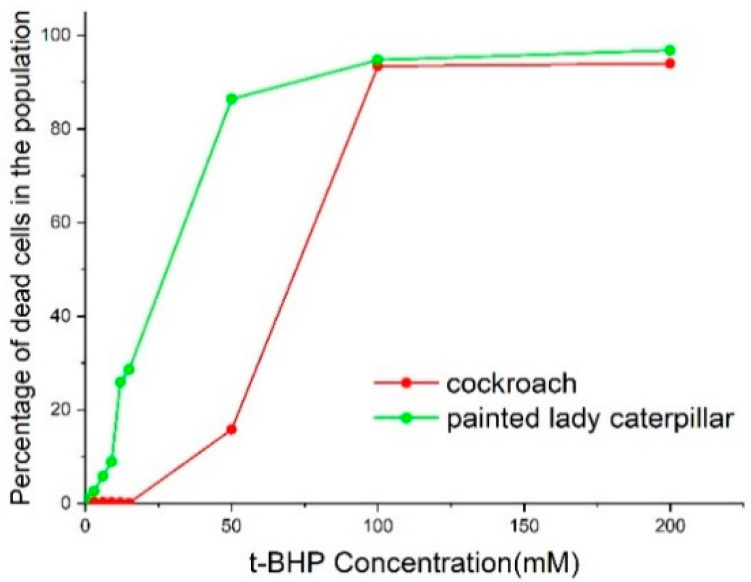
Percentage of the sum of apoptotic and dead cells in cockroach nymph and painted lady caterpillar midguts after six-hour exposure at eight concentrations of t-BHP.

**Figure 5 insects-14-00241-f005:**
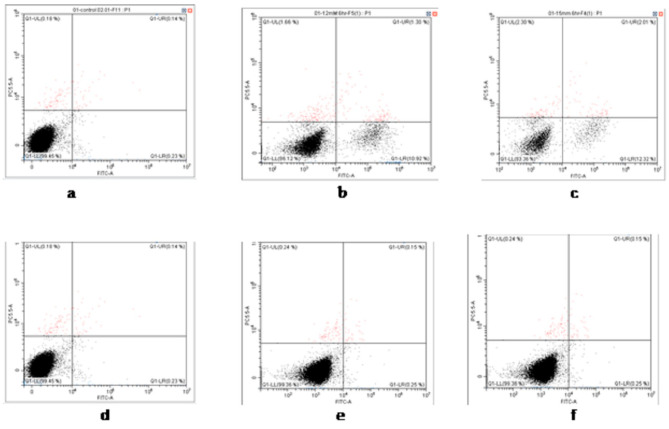
Examples of flow cytometry analysis of apoptosis of painted lady caterpillar cells (**a** = control, **b** = 12 mM t-BHP and **c** = 15 mM t-BHP) and Turkestan cockroach cells (**d** = control, **e** = 12 mM t-BHP and **f** = 15 mM t-BHP).

**Figure 6 insects-14-00241-f006:**
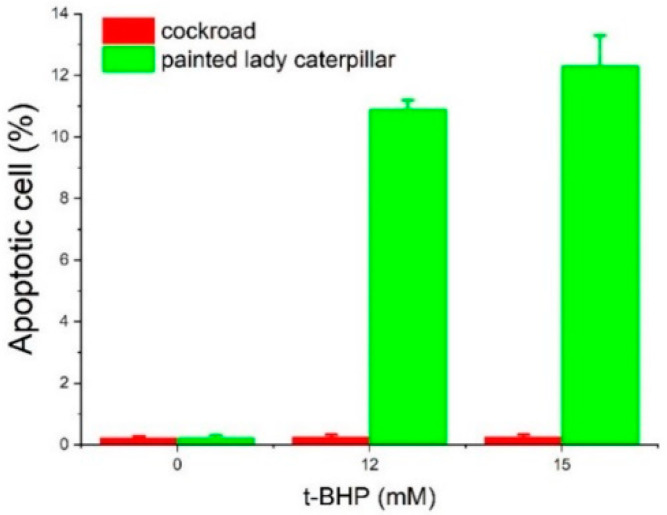
The fraction of apoptotic cells from painted lady caterpillars and cockroaches.

**Figure 7 insects-14-00241-f007:**
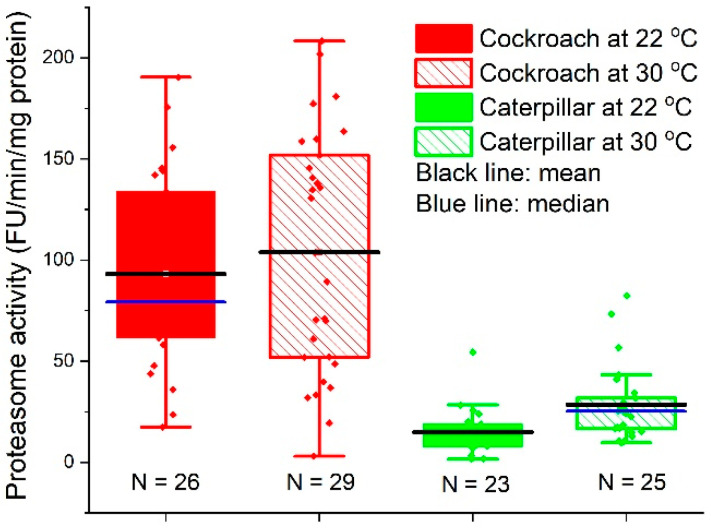
Activities of proteasome in midgut cells from cockroach nymphs and painted lady caterpillars that were reared at 22 and 30 °C. The assays were performed at 25 °C.

**Figure 8 insects-14-00241-f008:**
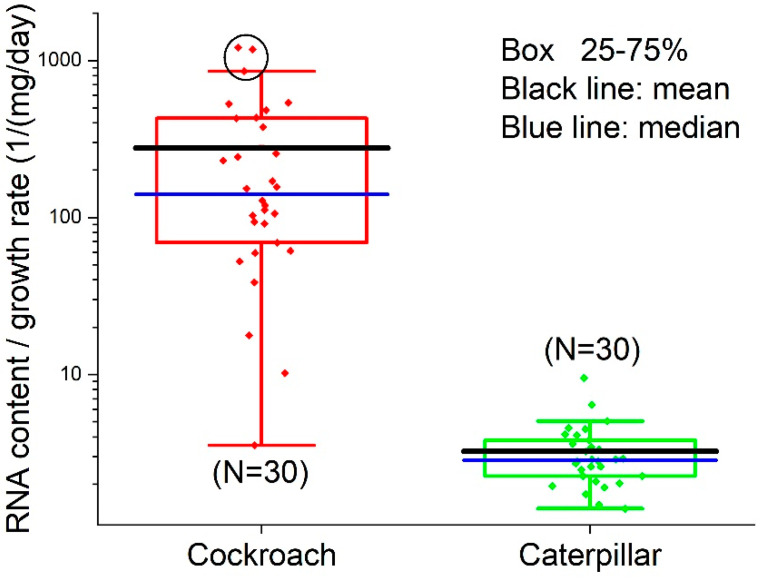
The RNA/growth ratio of cockroach and caterpillar midguts. Three outliers in cockroach dataset are circled. Note, the figure is in logarithm scale, so the points at the bottom of the cockroach panel are not identified by OriginLab as outliers.

## Data Availability

The data presented in this study are available in this article.

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
