# Peer review of "Link between Energy Investment in Biosynthesis and Proteostasis: Testing the Cost–Quality Hypothesis in Insects"

_insects, 2023, doi:10.3390/insects14030241_

Round 1

Reviewer 1 Report

The topic of this manuscript is very novel and interesting. The authors have put forth a hypothesis about the potential for proteostasis to contribute to species-level differences in growth rate and provide some evidence to support their claim. They use two species that vary in growth rate and Em, painted lady caterpillars and the Turkestan cockroach. They show that the midgut cells of painted ladies are more resistant to oxidative stress than cockroach cells. They also show that proteosome activity is higher in cockroach midgut cells. They also attempt to show that the RNA-to-growth rate ratio is higher for cockroaches than painted ladies (there are some issues with these data, however). These results largely support their hypothesis.

Despite this, there are several issues with the manuscript in it's current state. The writing needs a lot of work. There are issues with verb tense, missing articles, awkward sentences, wrong paragraph structure, etc. There is also a lot background information missing in the Introduction, which is needed to outline the importance of the question and the other work that has been done on the topic. There are details missing in the Methods. The Results text could be more detailed, although the figures are good. The RNA, protein, and RNA/growth data need to be re-thought, as the standard errors for many of those values are very higher (sometimes higher than the average value) making them difficult to interpret. I have placed more detailed comments in the edited manuscript.

Improving the writing and framing the question in a more useful context will make the manuscript much clearer.

Reviewer 2 Report

Link between energy investment in biosynthesis and proteostasis: Test the cost-quality hypothesis in insects.

Ironimi, et al.  Insects 2023

The cost-quality hypothesis is a physiological explanation for the measured differences in the amount of energy an organism expends to acquire a specific amount of biomass (dry) (Em) (KJ/gdm).  Further, the cost of biosynthesis is related to the quality of the biomass.  The Authors focus on the costs associated with proteostasis, or creating and maintaining the protein biomass, as the main factor determining Em.

The Em of a cockroach species is 10X that of a caterpillar.  Thus, a nymphal cockroach spends more energy creating an equivalent amount of biomass as the caterpillar and should have ‘higher quality’ biomass.  The Authors use these species in a test of the cost-quality hypothesis.

Figure 1. Are there examples of mammals or other insects the authors could include? I tend to think that fish, reptiles and amphibians may be more similar in biomass protein:lipid ratio than hemimetabolous and holometabolous insects, since the latter two groups differ so much in apparent biomass relative protein and lipid content. I understand that the Authors have published work showing tissue body composition does not strongly explain the differences in growth rate between the two species, but it seems that a recapitulation of those results should be mentioned in the current study to alleviate readers from their concerns.

The Authors test three predictions of the cost-quality hypothesis.  First, they test the quality of the proteosome by exposing cells from a cockroach and a caterpillar to BTH, an inducer of oxidative stress (by attacking phospholipids). The prediction is that the ‘higher quality’ detoxification machinery of the cockroach would better protect cells.

According to the Authors’ previous publication, the cockroach has less protein (and more lipids) relative to the caterpillar. Can the Authors comment on this difference in the current manuscript?

Line 54: The Authors state the difference in Em between a cockroach and a caterpillar can be understood in terms of their hemi and holometabolous life histories.  In the latter the authors argue that a large proportion of biomass created is energy storage for reproduction. This implies glycogen and lipid storage, neither of which would require energy to maintain. Can the Authors comment on the differences in body tissue composition?

Line 105: The biomass range is similar between cockroach and caterpillar, but the type of tissues comprising the biomass differ between the two.  Can you comment on this?

Minor comments:

Line 151: midguts were, rather than midgut were

Line 159: What tissue was used? Earlier in the paragraph, the whole body minus the midgut was used, now the thorax? Please clarify.

Line 161: inhibitor

Line 175: sentence fragment should be removed.

Line 188: ‘were cleaned’ rather than ‘will have to be cleaned’

Line 201: when were the midguts dried? It was unclear where the dried tissues came from originally.

Line 200: The data analysis section should contain the methods used to analyze the other data types, not just the RNA work.  Were the statistical analyses used justified?

Line 244: An ANOVA is used to analyze the proteasome data, but the data are graphically represented as medians.  Why this difference?  Was using an ANOVA justified for this analysis?

Figure 8: Same issue as above.  Was using an ANOVA justified? If so, why were not means (and st. err or st. dev) used instead of medians and quartiles?
